# Previously Undescribed Gross *HACE1* Deletions as a Cause of Autosomal Recessive Spastic Paraplegia

**DOI:** 10.3390/genes13122186

**Published:** 2022-11-23

**Authors:** Valeriia A. Kovalskaia, Victoriia V. Zabnenkova, Marina S. Petukhova, Zhanna G. Markova, Vyacheslav Yu. Tabakov, Oxana P. Ryzhkova

**Affiliations:** Research Centre for Medical Genetics, 115522 Moscow, Russia

**Keywords:** *HACE1*, spastic paraplegia, neurodevelopmental syndrome

## Abstract

Spastic paraplegia and psychomotor retardation with or without seizures (SPPRS, OMIM 616756) is a rare genetic disease caused by biallelic pathogenic variants in the *HACE1* gene. Originally, these mutations have been reported to be implicated in tumor predisposition. Nonetheless, via whole exome sequencing in 2015, *HACE1* mutations were suggested to be the cause of a new autosomal recessive neurodevelopmental disorder, which is characterized by spasticity, muscular hypotonia, and intellectual disability. To date, 14 *HACE1* pathogenic variants have been described; these variants have a loss-of-function effect that leads to clinical presentations with variable severities. However, gross deletions in the *HACE1* gene have not yet been mentioned as a cause of spastic paraplegia. Here, we report a clinical case involving a 2-year-old male presenting with spasticity, mainly affecting the lower limbs, and developmental delay. Exome sequencing, chromosomal microarray analysis, and mRNA analysis were used to identify the causative gene. We revealed that the clinical findings were due to previously undescribed *HACE1* biallelic deletions. We identified the deletion of exon 7: c.(534+1_535-1)_(617+1_618-1)del (NM_020771.4) and the gross deletion in the 6q16.3 locus, which affected the entire *HACE1* gene: g.105018931_105337494del, (GRCh37). A comprehensive diagnostic approach for the patients with originally homozygous mutations in *HACE1* is required since false homozygosity results are possible. More than 80% of the described mutations were reported to be homozygous. Initial hemizygosity is hard to detect by quantitative methods, and this may challenge molecular diagnostic identification in patients with spastic paraplegia.

## 1. Introduction

Spastic paraplegia and psychomotor retardation with or without seizures (SPPRS, OMIM 616756,) are rare genetic conditions caused by biallelic pathogenic variants in the *HACE1* gene. This gene was initially described as a cause of developmental delay in eight patients from two non-related families in 2015 via whole-exome sequencing [1]. Originally, the *HACE1* gene was only considered as an onco-suppressor, and the mutations were reported to be implicated in tumor predisposition [2]. It was reported to be downregulated in patients with sporadic Wilms tumors and translocations affecting the 6q21 locus: t(6;15)(q21;q21), t(5;6)(q21;q21), and t(2;6)(q35;q21) [2,3,4,5].

*HACE1* (HECT domain and Ankyrin repeat Containing E3-ubiquitin-protein ligase 1) encodes a 130 kDa protein with subcellular localization; it is implicated in the ubiquitination and proteasomal degradation of target proteins [6] and is highly expressed in various brain regions [7]. It controls cell migration by the regulation of cellular GTPases, such as Rac1, which are involved in the development of brain structures [8,9] and are dysregulated in patients with some neurodevelopmental disorders, including mental retardation and atypical synaptic plasticity [10,11].

Nagy V. also proved that *HACE1* is implicated in the formation of neurodevelopmental disorders by using *HACE1* knock-out mice, which displayed many clinical features of SPPRS [12]. The mice had enlarged ventricles, hypoplastic corpus callosum as well as learning impairment and motion difficulties. He reported that *HACE1* deficiency results in the decreased number of synaptic puncta and in long-term potentiation in the hippocampus.

To date, 14 *HACE1* pathogenic variants have been described; these are loss-of-function variants that lead to clinical presentations with variable severities (Table 1). However, gross deletions in this gene have not yet been reported as a cause of spastic paraplegia and psychomotor retardation with or without seizures (OMIM 616756) [13]. In the majority of cases, the clinical findings on this disorder include psychomotor retardation, communicative difficulties, developmental delay, epilepsy, diffuse hypotonia, spasticity in the lower and occasionally in the upper limbs, ataxia, and impaired gait [13]. Affected individuals may also present with genital hypoplasia, congenital macrocephaly (91–99 percentile), increased body mass, ophthalmic findings (myopia, strabismus, retinal dystrophy), and sensorineural hearing loss [1]. MRI findings include ventriculomegaly, global cerebral cortex atrophy or frontal/temporal lobe atrophy, and corpus callosum hypoplasia [1,13].

In this study, we report a clinical case involving a male Russian patient born in 2018 and presenting with severe spasticity, mainly affecting the lower limbs, developmental delay, diffuse hypotonia, mild dysmorphic features, and brain MRI changes without seizures. These clinical symptoms were caused by biallelic gross *HACE1* deletions and these types of mutations have not yet been described in *HACE1* gene. We detected a deletion of exon 7 c.(534+1_535-1)_(617+1_618-1)del (NM_020771.4) and a gross deletion in the 6q16.3 locus spanning the entire *HACE1* gene: g.105018931_105337494del, (GRCh37).

## 2. Materials and Methods

### 2.1. Patient Case

The proband—a male 2.8-year-old Russian patient (Figure 1 and Figure 2)—was examined by a geneticist in the Research Centre for Medical Genetics (Moscow, Russia) in the presence of his parents. The marriage of the proband’s parents was not consanguineous (Figure 3). The blood samples and skin fibroblasts of the proband and his parents were obtained with informed consent for molecular genetic analysis and anonymous participation in the scientific research.

### 2.2. Diagnostics

Clinical exome sequencing (CES) was carried out for the proband (Figure 4). The patient’s DNA was analyzed on a next-generation Illumina NextSeq 500 sequencer using the paired-end reading (2 × 75 bp) approach. The probe preparation was based on the selective capture of DNA fragments representing the coding regions of 6640 genes (Appendix A), which had been described as clinically significant, using the SeqCap EZ Prime Choice XL Probes (Roche Sequencing Solutions, Pleasanton, CA, USA). The average coverage depth was 86.4×, and the width of ×10 coverage was at 98.8986% and of ×20 coverage was at 94.0322%. The sequencing data were processed using the standard automated data analysis algorithm provided by Illumina (https://basespace.illumina.com (accessed on 21 September 2022). To evaluate the population frequencies of the detected variants, the data from the 1000 genomes project, ESP6500, and gnomAD (v.2.1.1) were used.

### 2.3. Mutation Verification

The deletion of exon 7 of the *HACE1* gene was verified using mRNA analysis (Figure 5 and Figure 6). Fibroblasts were obtained from the patients (proband and his parents) by means of forearm skin biopsy, in accordance with the current standards. The obtained fibroblasts were cultivated in a DMEM medium supplied by 15% fetal bovine serum at 37 °C in an incubator with 5% CO_2_. After a time, the fibroblasts from the primary cell cultures were used to extract the total RNA using the QIAamp^®^ RNA Blood Mini Kit (Qiagen, Hilden, Germany). The quality of the total RNA extracted was evaluated by means of electrophoresis in agarose gel and the spectrophotometric estimation of the 260/280 ratio during the measurement of the total RNA concentration. The reverse transcription of the total RNA samples was performed using the QuantiTect Reverse Transcription Kit (Qiagen). Standard PCR for cDNA samples of the proband and his parents as well as for reference samples (cDNA samples of non-affected subjects) was carried out using primers flanking introns 4 and 9 of the *HACE1* gene (NM_020771.4). The primer sequences were 4F: TTTGGCAGCAAGAAATGGGC and 10R: ATGCTCCAGAACTTGCCGTA. In addition, chromosomal microarray analysis was conducted for the proband using Affimetrix CytoScan HD oligonucleotide microarrays with a 25 kb resolution in order to identify the origin of the detected deletion in the *HACE1* gene.

## 3. Results

### 3.1. Patient Examination

The proband is a male patient (Figure 2 and Figure 3), examined twice at the age of 2 years and 8 months and at 3 years and 5 months. He was born to non-consanguineous, healthy parents and had no affected relatives (1st pregnancy, 1st delivery at full-term). The birth weight was 3100 g (between the 25th and 50th percentile), length was 52 cm (between the 75th and 90th percentile), and head circumference was 34 cm (between the 25th and 50th percentile). The psychomotor retardation was remarkable; by the first examination, the proband was determined to have a speech delay and had a limited perception of the speech. He was also unable to sit independently and had never walked, although he was able to hold his head up and roll over. During the physical examination, he was noted to have insufficient weight gain as well. His physical characteristics by the first examination were as follows: height—81 cm, weight—10 kg, low level of physical development, head circumference—46 cm. Additional clinical findings include mild facial dysmorphic features (coarse facies, low anterior hairline, thick eyebrows, long eyelashes, convergent strabismus, long philtrum, a short neck, nipple hypertelorism, transverse palmar crease, and joint hypermobility). Previously, the diffuse muscular hypotonia had been reported; nevertheless, the second neurological examination showed spastic tetraparesis and exaggerated deep tendon reflexes in all four limbs (more apparent in the lower limbs). Cranial MRI revealed moderate ventriculomegaly and severe cortex atrophy. The urinary organic acids profile was within normal limits, and 3-metylglutaconic acid was not particularly elevated.

### 3.2. Clinical Exome Sequencing

The analysis of clinical exome sequencing results did not reveal any pathogenic or likely pathogenic variants. However, the CNV analysis showed the possible deletion of exon 7 of the *HACE1* gene (NM_020771.4) in a homozygous or hemizygous state (Figure 4).

### 3.3. mRNA Analysis

PCR was performed using primers that flank introns 4 and 9 of the *HACE1* gene and the cDNA samples from the proband (F1.1), his mother (F1.2), and father (F1.3) as well as reference samples (R1–R5). The generated amplicons were detected by means of electrophoresis in polyacrylamide gel (I, II, III, IV). Each band was analyzed via Sanger sequencing (Figure 5 and Figure 6). The analysis confirmed the deletion of exon 7 on both alleles in the proband. His father was also likely to carry the deletion of exon 7; however, in the cDNA of the proband’s mother, who was expected to be a carrier of exon 7 deletion as well, it was not detected. Considering the obtained data, we suggested either the presence of another gross deletion involving exon 7 of the *HACE1* gene on the second allele, a uniparental disomy, or a de novo deletion of exon 7 on the second allele. To confirm one of these suggestions, we carried out a chromosomal microarray analysis.

### 3.4. Chromosomal Microarray Analysis

The chromosomal microarray analysis results showed a microdeletion of 319 kb from the long arm of chromosome 6, which spanned the entire sequence of the *HACE1* gene: arr(GRCh37) 6q16.3(105018931_105337494)x1 (Figure 7). Variants with similar coordinates were not registered in the general population DGV database or in the clinical DECIPHER database. Loss of heterozygosity in imprinted regions were absent. The total length of the regions with loss of heterozygosity corresponds to the populational values (0.14%). Thus, the patient’s genotype included the variants c.(534+1_535-1)_(617+1_618-1)del (NM_020771.4) and g.105018931_105337494del, del (NM_020771.4) (GRCh37) in a compound heterozygous state.

## 4. Discussion

Spastic paraplegia and psychomotor retardation with or without seizures (OMIM 616756) is an autosomal recessive disorder caused by the presence of two mutations in the *HACE1* gene in trans. To date, the pathogenic *HACE1* variants are limited to 14 LoF mutations (Figure 8), as reported in the HGMD Professional Database v.2022.1; no gross deletions have been described as the cause of this neurodevelopmental disorder. With this study, we are the first to have detected two pathogenic, biallelic, gross deletions in an affected male: the deletion of exon 7 of the *HACE1* gene and a gross deletion in locus 6q16.3 (319 kb) that spans the entire *HACE1* gene.

The mRNA analysis showed the absence of the transcripts containing exon 7 in the proband and suggests the paternal inheritance of this mutation (Figure 5). Unfortunately, certain *HACE1* transcripts do not contain the sequence of exon 7 (isoform d) nor exons 6 and 7 (isoform h), which are normally expressed in cultivated fibroblasts with a proper protein formation: 741 aa and 715 aa, respectively (Appendix A). Therefore, amplicons without exon 7 were registered in all samples, including the references (Figure 5); however, the canonical transcript was present in all healthy individuals but was absent in the proband. Moreover, the proband’s deletion of the exon 7 led to the frameshift and did not allow for a full-sized protein synthesis due to NMD. (Figure 9). The increase in the signal intensity of bands II and IV in the father’s material (F1.3) allowed us to assume the paternal inheritance of the *HACE1* exon 7 deletion (Figure 5). Considering that CMA was carried out only for the proband, we could not establish whether the deletion of the entire *HACE1* gene was maternally inherited or de novo; however, the absence of exon 7 on both alleles allowed us to confirm the trans position of the detected mutations.

According to the DECIPHER database [20], the vast majority of the described causative mutations, and the deletion of exon 7 in particular, in the *HACE1* gene are located in the NMD region (nucleotides 101 to 2574), which, as predicted, leads to the non-translation of a protein, truncated or not, due to nonsense-mediated decay. Thus, the unified etiopathogenetic mechanism for the most described variants is the complete absence of the HACE1 protein.

As shown in Table 1, the majority (10 out of 14) of biallelic *HACE1* variants are homozygous. However, as noted in the current study, *HACE1* deletions contribute to the structure and diversity of causative variants in the *HACE1* gene, and false homozygosity results are possible. Thus, family segregation analysis may be required for proper risk assessment.

The patient in this study, similar to previously described cases, had profound hypotonia and mental retardation. Spastic bilateral paraplegia was a key symptom, however, the spasticity in the upper limbs was less pronounced. In spite of the fact that the term “spastic paraplegia” is present in the disease title, this symptom was not present in all described patients. In 2 cases out of 14, dystonia was noted, and in 4 cases, spastic paraplegia was not described at all. The proband in the current clinical case did not have any signs of epilepsy, which was described in the majority of cases (7 out of 10). According to the data, this disorder is characterized by myoclonic epilepsy without generalized seizures as well as by ocular impairments, including myopia, strabismus, and retinopathy. At least one of these symptoms has been detected in every clinically examined patient. In two cases out of six, sensorineural hearing loss was reported; however, it was not detected in our patient. In all cases, by means of instrumental investigations, brain damage was noted. The most common symptoms were hypoplastic corpus callosum (seven cases out of nine), cerebral atrophy (four out of nine), and enlarged ventricles/ventriculomegaly (four out of nine). The latter two were detected in our patient. In almost every case, certain dysmorphic features were noted. In the current case, the facial phenotype included coarse facial features, low anterior hairline, thick eyebrows, and long eyelashes, and all of them have not been described in patients with a similar disorder.

It is worth noting that despite the limited number of patients with described pathogenic variants in the *HACE1* gene, we can assume the presence of clinical heterogeneity, not only between non-related patients but even within one family. 

Holsteins R. reported that one male out of four affected siblings had no epilepsy, while all the rest of the affected subjects had myoclonic or tonic–clonic epilepsy manifesting at various ages with the homozygous p.R219* mutation. Their ability to walk also differed dramatically. The ophthalmic findings for other family members with the p.R748*/p.P674Ffs*5 genotype ranged from none to convergent strabismus, bilateral myopia, and retinal dystrophy [1]. This contributes to the necessity for a thorough examination of all members of the affected family.

## 5. Conclusions

The obtained results emphasize the necessity for a comprehensive examination of patients with homozygous variants in the *HACE1* gene, as hemizygosity, which is complicated or impossible to detect with qualitative molecular genetic methods, may prevail. An examination of the parents and the confirmation of their carrier status would allow us to combat these challenges. In addition, clinical variability may present among family members with biallelic mutations of the *HACE1* gene; thus, precise examination is highly recommended.

## Figures and Tables

**Figure 1 genes-13-02186-f001:**
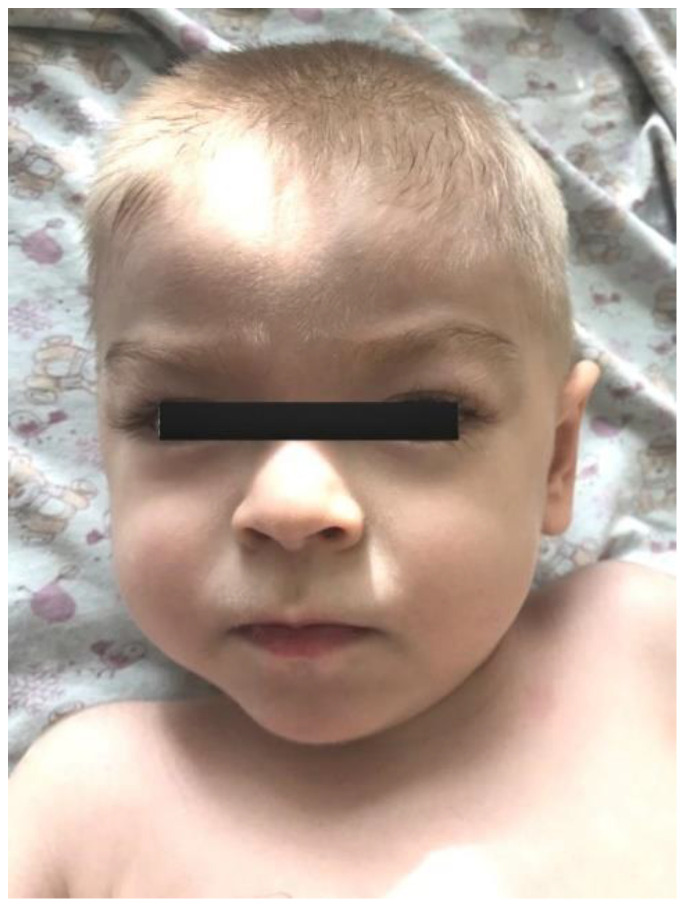
The proband’s facial phenotype.

**Figure 2 genes-13-02186-f002:**
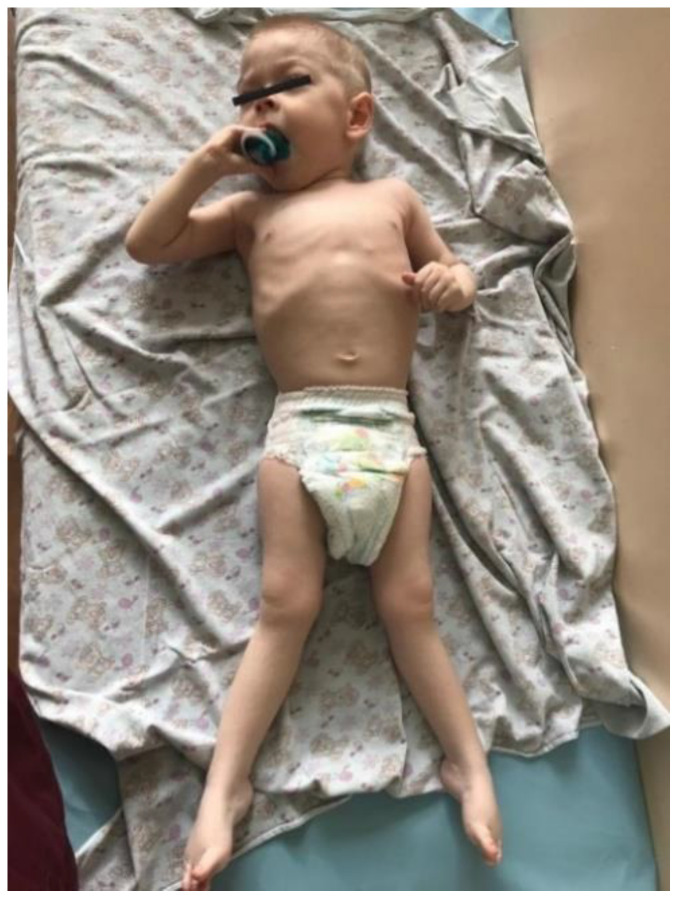
Spastic paraplegia in the proband’s lower limbs.

**Figure 3 genes-13-02186-f003:**
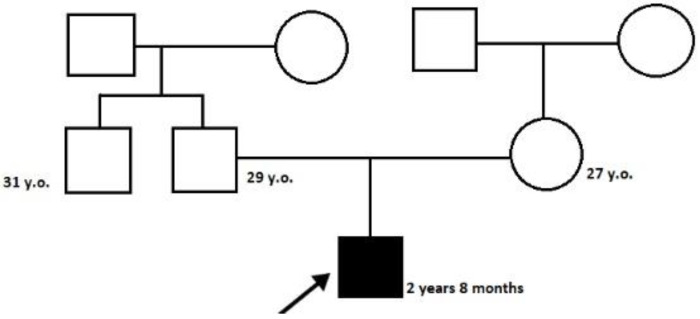
The pedigree of the proband.

**Figure 4 genes-13-02186-f004:**
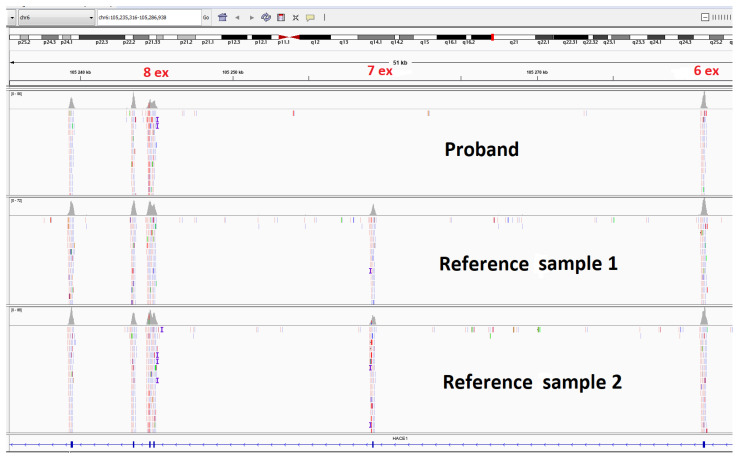
Molecular genetic testing of the patient by clinical exome sequencing and visualization of the homo/hemizygous deletion of *HACE1* exon 7.

**Figure 5 genes-13-02186-f005:**
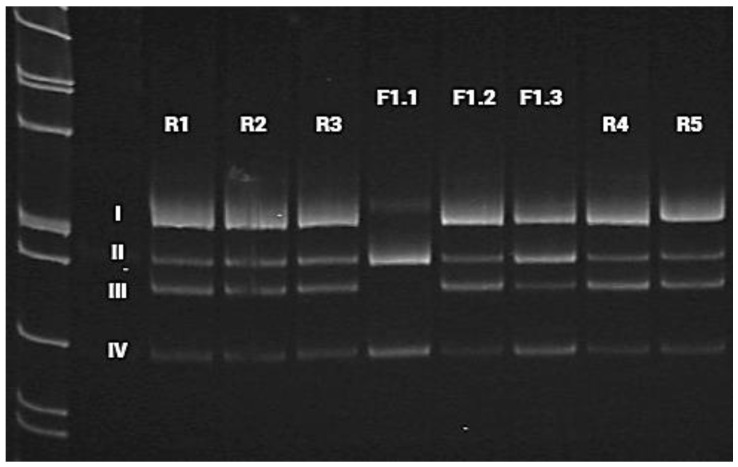
Visualization of cDNA amplification products for the proband (F1.1), his parents (F1.2 and F1.3), and reference samples (R1–R5), with primers flanking introns 4 and 9 of the *HACE1* gene. Band I (529 bp) includes the sequences of exons 5, 6, 7, 8, and 9 of the *HACE1* gene, which corresponds to the canonical protein-coding transcript 1 (isoform a). This amplicon was absent in the proband (F1.1). Band II (446 bp) includes the sequences of exons 5, 6, 8, and 9 of the *HACE1* gene without exon 7 (isoform d). Band III (397 bp) corresponds to protein isoforms b and e, whose transcripts do not contain the sequence of *HACE1* exon 6, but contain exon 7. Band IV (314 bp) does not include the sequences of exons 6 and 7 of the *HACE1* gene, which corresponds to the transcript of *HACE1* isoform h. Bands I and III were not registered in the proband (F1.1). In the proband’s father (F1.3), we detected an increase in signal intensity for bands II and IV as compared to the reference samples (R1–R5) and the maternal sample (F1.2).

**Figure 6 genes-13-02186-f006:**
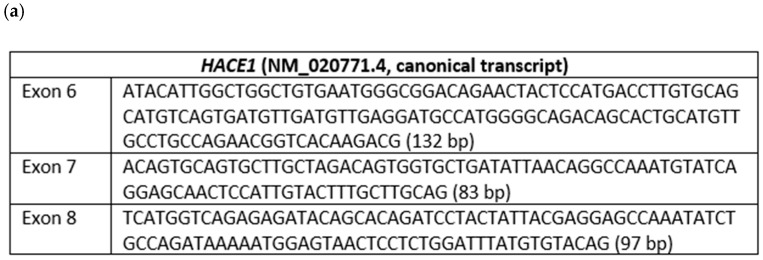
(**a**) Nucleotide sequences and lengths of exons 6, 7, and 8 of the *HACE1* gene (canonical transcript); (**b**) Sanger sequencing validation of the homozygous/hemizygous deletion of exon 7 of the *HACE1* gene in the proband; (**c**,**d**) Sanger sequencing of the patient’s mother’s cDNA, validating the presence of exons 6, 7, and 8 of the *HACE1* gene; (**e**,**f**) Sanger sequencing of patient’s father’s cDNA, validating the presence of exons 6, 7, and 8 of the *HACE1* gene.

**Figure 7 genes-13-02186-f007:**
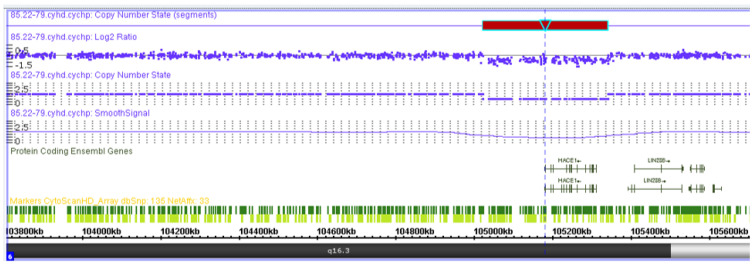
Visualization of the heterozygous deletion of the entire *HACE1* gene in the patient by means of chromosomal microarray analysis.

**Figure 8 genes-13-02186-f008:**
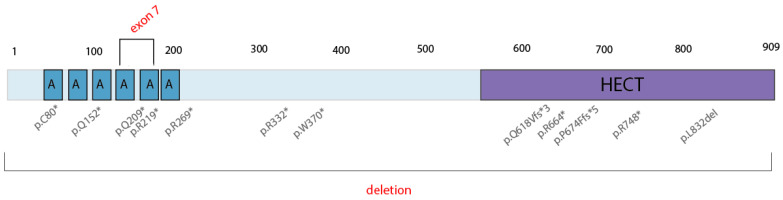
Described pathogenic variants in the *HACE1* gene. * = nonsense mutation.

**Figure 9 genes-13-02186-f009:**
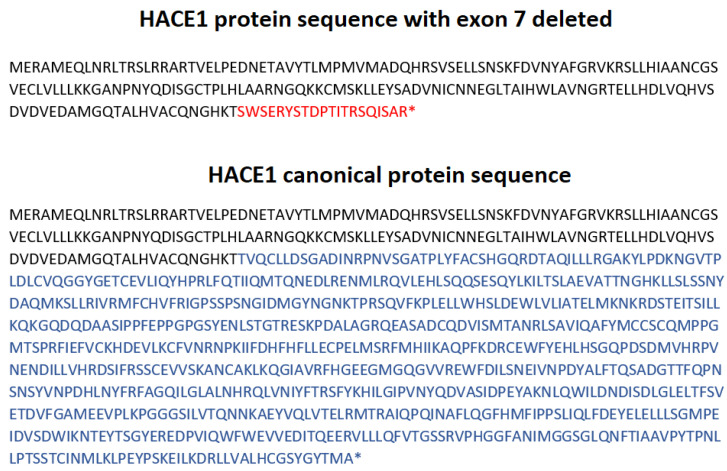
HACE1 protein sequence, as long as exon 7 is deleted, and HACE1 canonical protein sequence (NM_020771.4)**.** *= termination of the protein synthesis.

**Table 1 genes-13-02186-t001:** Clinical of the affected individuals with biallelic *HACE1* mutations.

Sex, Age, Mutation Types	Origin	Mobility	Hypotonia Reported	Developmental Delay	Spacticity (Lower Limbs)	Spacticity (Upper Limbs)	Epilepsy, Seizures	Ophthalmic Findings	Hearing	Instrumental Investigations	Other Features	Reference
Family (1F, 4M): p.R219*/p.R219*	Pakistan (consanguineous marriage)	Never walked (almost all affected subjects)	Yes	Yes	Bilateral spasticity	Dystonic posturing/ None	Myoclonic or tonic–clonic epilepsy (almost all affected subjects)	Divergent strabismus, myopia, retinal dystrophy	Bilateral sensorineural loss/ Normal	CT scan: generalized cerebral atrophy, ventricular dilatation EEG: consistent with myoclonic epilepsy	Hypogenitalism, kyphoscoliosis overweight	[1]
Family (1F, 2M): p.R748*/p.P674Ffs*5	Germany (non-consanguineous marriage)	Unsteady, broad-based ataxic gait	Yes	Yes	Bilateral spasticity	Dystonic posturing/ None	Myoclonic seizures, focal epilepsy/None	Convergent strabismus, bilateral myopia	Normal	MRI: enlarged ventricles, hypoplastic corpus callosum	Mild talipes equinovarus, lumbar lordosis	[1]
Family (1M, 1F): p.R332*/p.R332*	Turkey (consanguineous marriage)	Not at 6 months	Yes	Developmental delay	Not reported	Not reported	No	Strabismus	Normal	n/a	Small feet, enlarged head circumference, inverted and widely spaced nipples, facial dysmorphic features	[12]
Family (1M): p.Q209*/p.Q209*	Saudi Arabia (consanguineous marriage)	Inability to sit and walk at 5 years	Yes	Severe intellectual disability, Developmental delay	Not reported	Not reported	No	Retinal dystrophy	Bilateral sensorineural loss	MRI: microcephaly, hypoplastic corpus callosum, brainstem abnormality, small sella with ectopic neurohypophysis, and mild ventriculomegaly	Mild facial dysmorphia, skeletal abnormalities, ulnar deviation of the wrists and small feet, bruxism	[12]
Family (1M): p.Q152*/ p.Q618Vfs*3	n/a	n/a	Yes	Yes	Yes	n/a	Yes	n/a	Not reported	MRI: hypoplastic corpus callosum, lesions	Primary microcephaly	[14]
Family (1F): p.R269*/p.R269*	n/a (consanguineous marriage)	n/a	Yes	Yes	Not reported	Not reported	Yes	n/a	Not reported	MRI: hypoplastic corpus callosum, lesions	n/a	[14]
Family (1M): p.C80*/p.C80*	n/a (consanguineous marriage)	n/a	Yes	Yes	Not reported	Not reported	No	n/a	Not reported	MRI: hypoplastic corpus callosum, lesions	n/a	[14]
Family (1M): p.C80*/p.C80*	Pakistan (non-consanguineous marriage)	By 10 years was still not able to walk independently		Severe intellectual disability with greater impairment in the area of language	Paresis of the lower extremities with rigidity and exalted osteotendinous reflexes	Not reported	Not reported	n/a	n/a	MRI: diffuse cortical atrophy and an arachnoid cyst in the right temporal lobe	3-methylglutaconic aciduria	[15]
Family (2F, 1M): p.L832del/p.L832del	n/a (consanguineous marriage)	n/a	Yes	Yes	Spasticity/ dystonia	n/a	Seizures	n/a	Not reported	MRI: hypoplastic corpus callosum, cerebral atrophy	Primary microcephaly	[14]
Family (1M): p.W370*/p.W370*	India (consanguineous marriage)	Broad-based, crouched gait, no ataxia or fasciculations	Yes	Intellectual disability	Hypertonia, exaggerated deep tendon reflexes	Hypertonia, exaggerated deep tendon reflexes	Myoclonic seizures in limbs, no generalized seizures	Iris heterochromia, normal vision	Normal	MRI: hypoplastic corpus callosum, lesions	Hypopigmented body hair, proximal femoral focal dysplasia	[16]
Family (1F): p.R664*/p.R664*	n/a (consanguineous marriage)	n/a	n/a	Yes	Yes	Not reported	n/a	n/a	Not reported	n/a	n/a	[17]
Family (2F): c.402+5G>A/c.402+5G>A	Syria (consanguineous marriage)	Ataxic gait	Yes	Severe intellectual disability	n/a	n/a	Not reported	Not reported	Not reported	n/a	Recurrent infections	[18]
Family (2F + 1M): c.2212-1G>A/c.2212-1G>A	n/a (consanguineous marriage)	Motor developmental delay, regression	Yes	Intellectual disability	Dystonia	n/a	Seizures with abnormal movements	Eye movement deficit, horizontal	Not reported	n/a	Broad forehead and hypertelorism	[19]
Family (1M): ex 7 del/entire *HACE1* deletion	Russian (non-consanguineous marriage)	Never walked	Yes	Intellectual disability with greater speech impairment	Yes (more severe than in upper limbs)	Yes	No	n/a	Normal	MRI: enlarged ventricles, cerebral atrophy	Normal urinary levels of 3-metylglutaconic acid, coarse face, mild hyperthrichosis, long eyelashes	This study

n/a—Not available.

## Data Availability

Raw data are available from the corresponding author upon request.

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
