# Peer review of "Previously Undescribed Gross HACE1 Deletions as a Cause of Autosomal Recessive Spastic Paraplegia"

_genes, 2022, doi:10.3390/genes13122186_

Round 1

Reviewer 1 Report

The manuscript described a clinical case with a very rare exon 7 deletion in HACE1 plus gross deletion involving HACE1 which resulted in biallelic loss-of-function of HACE1 that cause potential HACE1-related recessive disorder. The clinical features of the patient are much consistent with previous reports on spastic bilateral paraplegia, server cortex atrophy by MRI, and developmental delay but without epilepsy.   

Some comments:

1. In the OMIM and monarch database, it had been characterized as a recessive neurodevelopmental syndrome called spastic paraplegia and psychomotor retardation with or without seizures caused by HACE1 loss-function mutations. However, the gene-disease clinical validation curation had not been well performed. So the gene function evidence of HACE1 and the phenotype of HACE1 deficiency in an animal model may be necessary to discuss.  

2. Please make sure that the nomenclature of the exon 7 deletion of HACE1 in this report is correct, especially about the deletion of nucleotides crossing an exon border (HGVS). Please also enclose the chromosomal location for this exon7 deletion.

3. The resolution of all figures in this manuscript is too bad and vague. Even it is difficult to figure out the mutant base pair by the Sanger sequence. 

4. The pathogenicity of single exon deletion had been presented using ACMG/AMP interpretation guidelines (Genetics in Medicine 2019). Exon 7 deletion in HACE1 in this case can be applied to this guideline. According to the interpretation guideline, whether exon 7 deletion leads to a null allele would be the impact of the reading frame which results in NMD. Did exon 7 deletion in HACE1 result in NMD (nonsense-mediated decay)? 

5. I suggested having a primer-location design scheme together with figure5. It is too difficult to understand the isoform-specific primers. Confused. In the discussion, it is mentioned there were some isoforms (d, h) of HACE1 which did not have exon7, but “the canonical transcript is present in all healthy individuals but is absent in the proband” in this article. Is there any evidence to support that the isoforms (d, h) are not active in affected tissue or canonical transcript is the tissue-specific main transcript?

6. Line 137-138, “….. to be a heterozygous carrier of this deletion; …….to have the same result” need to point out which deletion they carried because there are two deletions here. 

Author Response

Dear Reviewer 1,

Thank you so much for your worthy comments!
We have edited our manuscript in accordance with them.

  1. In the OMIM and monarch database, it had been characterized as a recessive neurodevelopmental syndrome called spastic paraplegia and psychomotor retardation with or without seizures caused by HACE1 loss-function mutations. However, the gene-disease clinical validation curation had not been well performed. So the gene function evidence of HACE1 and the phenotype of HACE1 deficiency in an animal model may be necessary to discuss.  

Animal model was added.
Nagy V. also proved, that HACE1 is implicated in neurodevelopmental disorder formation using Hace1 knock-out mice, that displayed many clinical features of SPPRS [13]. They had enlarged ventricles, hypoplastic corpus callosum, as well as learning impairment and motion difficulties. He reported that HACE1 deficiency results in decreced synaptic puncta number and long-term potentiation in the hippocampus. (lines 43-47).

  1. Please make sure that the nomenclature of the exon 7 deletion of HACE1 in this report is correct, especially about the deletion of nucleotides crossing an exon border (HGVS). Please also enclose the chromosomal location for this exon7 deletion.

Due to the fact, that the exact intronic break points of the deletion were not identified, we can use only the nomenclature recommended by http://varnomen.hgvs.org/recommendations/DNA/variant/deletion/

Thus, we are aware, that the exon 7 is deleted thanks to mRNA analysis and exome sequencing, but intronic borders are still unknown. Thus, we are not sure, that chromosomal location is possible to add due to this obstacle.

  1. The resolution of all figures in this manuscript is too bad and vague. Even it is difficult to figure out the mutant base pair by the Sanger sequence. 

We have adjusted the quality of the figures, thank you

  1. The pathogenicity of single exon deletion had been presented using ACMG/AMP interpretation guidelines (Genetics in Medicine2019). Exon 7 deletion in HACE1 in this case can be applied to this guideline. According to the interpretation guideline, whether exon 7 deletion leads to a null allele would be the impact of the reading frame which results in NMD. Did exon 7 deletion in HACE1 result in NMD (nonsense-mediated decay)? 

The deletion of the exon 7 leads to the frameshift and further nonsense-mediated decay, so no protein is synthesized, even truncated. We have mentioned that (lines 220-221, lines 227-230) and added a Figure 9, that demonstrates it properly.

  1. I suggested having a primer-location design scheme together with figure5. It is too difficult to understand the isoform-specific primers. Confused. In the discussion, it is mentioned there were some isoforms (d, h) of HACE1 which did not have exon7, but “the canonical transcript is present in all healthy individuals but is absent in the proband” in this article. Is there any evidence to support that the isoforms (d, h) are not active in affected tissue or canonical transcript is the tissue-specific main transcript?

We didn`t use isoform-specific primers, so unfortunately, it`s not possible to add them to Figure 5, however, in order to facilitate Figure 5 perception and isoforms structures understanding, we have added a Supplement Material 2, that fairly well demonstrates the primers’ location and isoform-specific amplicons.

  1. Line 137-138, “….. to be a heterozygous carrier of this deletion; …….to have the same result” need to point out which deletion they carried because there are two deletions here. 

Corrected it for «His father was likely to carry out the deletion of exon 7; however, in cDNA of the pro-band’s mother, who was expected to be the carrier of exon 7 deletion as well, it was not detected».

Reviewer 2 Report

The article demonstrates the importance of confirming the segregation of the found mutations by NGS and of considering other techniques (such as RNA studies) to arrive at the correct diagnosis.

However, there are minor changes that need to be made.

1.            In the introductions section, in line 42, other involvements in mitochondria and autophagy of HACE1 have been described, so it should be added (reference Olatz Ugarteburu et al. 2020, journal of clinical medicine 9; 913, and add this reference).

2.            In the results section, when the results are explained they don’t refer to the figure.  For example, add in section 3.3, line 136: “figure 5 and 6” when corresponding; in section 3.4 , line 174 add “figure 7”.

 3.           In table 1, the patient described in the above-mentioned reference (Olatz Ugarteburu et al. 2020, journal of clinical medicine 9; 913) should be added.

4.            In line 205 to say that maybe the described homozygous mutations can be false is strong, usually in the publications of the cases are mentioned if the segregation studies have been done, please review the publications,  and comment maybe the patients whose segregation studies have not been done, and so in these cases, more studies should be performed. Maybe it could comment on how the utilization of other methodologies such as RNAseq can improve tax diagnosis and that sometimes exome study is not enough, as has been described.

5.            It has been described one patient with HACE1 mutations presents 3-methylglutaconic acid in urine. Were organic acids analyses done in your patient? It should be mentioned if the organic acid analysis was done in your patient and if they were normal or not, and discuss with the publication (Olatz Ugarteburu et al. 2020, journal of clinical medicine 9; 913).

Author Response

Dear Reviewer 2,

Thank you so much for your remarkable comments regarding our manuscript.

  1. In the introductions section, in line 42, other involvements in mitochondria and autophagy of HACE1 have been described, so it should be added (reference Olatz Ugarteburu et al. 2020, journal of clinical medicine 9; 913, and add this reference).

We have not mentioned HACE1 involvements in mitochondria and autophagy in introduction, however we find it relevant to cite Olatz Ugarteburu et al. 2020 in our article. Reference 19.

  1. In the results section, when the results are explained they don’t refer to the figure. For example, add in section 3.3, line 136: “figure 5 and 6” when corresponding; in section 3.4 , line 174 add “figure 7”.

References to the figures were added

  1. In table 1, the patient described in the above-mentioned reference (Olatz Ugarteburu et al. 2020, journal of clinical medicine 9; 913) should be added.

The patient`s clinical presentation, reported by Olatz Ugarteburu et al. 2020 was added

  1. In line 205 to say that maybe the described homozygous mutations can be false is strong, usually in the publications of the cases are mentioned if the segregation studies have been done, please review the publications, and comment maybe the patients whose segregation studies have not been done, and so in these cases, more studies should be performed. Maybe it could comment on how the utilization of other methodologies such as RNAseq can improve tax diagnosis and that sometimes exome study is not enough, as has been described.

False homozygous results are possible as long as family segregation analysis has not been performed. We`ve specified that (line 234-237).

«However, as noted in the current study, HACE1 deletions contribute to the structure and diversity of causative variants in the HACE1 gene and false homozygosity results are possible. Thus, family segregation analysis may be required for further risk assessment.»

  1. It has been described one patient with HACE1 mutations presents 3-methylglutaconic acid in urine. Were organic acids analyses done in your patient? It should be mentioned if the organic acid analysis was done in your patient and if they were normal or not, and discuss with the publication (Olatz Ugarteburu et al. 2020, journal of clinical medicine 9; 913).

Urinary organic acids profile was done for our patient. It was within normal limits. Added this information (line 121-123) + Table 1.
We cited Olatz Ugarteburu et al. 2020 (reference 19) as we found it relevant for our article.

Reviewer 3 Report

The manuscript entitled "Previously undescribed gross HACE1 deletions as a cause of autosomal recessive spastic paraplegia" is proposed to publish as an original paper. The authors report a 2-years-old patient with compound heterozygous microrearrangements: deletion of exon 7 of the HACE1 gene inherited from the father, and gross deletion encompassing the entire HACE1 gene inherited from the mother. The microrearrangements within the HACE1 gene have not been previously described. 

The novelty of the article is related to previously undescribed mutations - gross deletions in the HACE1 gene resulting the spastic paraplegia and psychomotor retardation with or without seizures. This phenotype is related to the HACE1 gene, however, up to date, only single nucleotide pathogenic variants have been described. 

The strengths of the manuscript are clinical data and comparison and characteristics of signs in patients with the HACE1 mutations. The table is a clear summary of clinical data.

The main objections are as follows:

1. The implication of the exon 7 deletion is not explained conscientiously. The absence of transcript in the proband suggests the mechanism of transcript destroying (nonsense-mediated decay mechanism may be implicated). Furthermore, conclusions about the father's exon 7 deletion in the HACE1, according to the cDNA amplification, are doubtful. cDNA amplification is not a quantitative method thus analysis of the polyacrylamide gel electrophoresis should not be used for that conclusion.

2. The impact of the presence of various HACE1 transcripts should be discussed in detail.    

3. English should be improved.

4. Low quality of the figures.

In conclusion, the manuscript reports new gross deletions in the HACE1 gene. It should be presented as a case report rather than an original paper.   For the original article, a more detailed transcript analysis should be presented. The authors performed a couple of genetic tests, however, the analysis of the results is insufficient.

Author Response

Dear Reviewer 3,

Thank you for your feedback, we have edited our manuscript in accordance with your comments and suggestions.

  1. The implication of the exon 7 deletion is not explained conscientiously. The absence of transcript in the proband suggests the mechanism of transcript destroying (nonsense-mediated decay mechanism may be implicated). Furthermore, conclusions about the father's exon 7 deletion in the HACE1, according to the cDNA amplification, are doubtful. cDNA amplification is not a quantitative method thus analysis of the polyacrylamide gel electrophoresis should not be used for that conclusion.

The deletion of the exon 7 leads to the frameshift and further nonsense-mediated decay, so no protein is synthesized, even truncated. We have mentioned that (lines 220-221, lines 227-230) and added Figure 9, that demonstrates it properly.

You are right, that we are not able to assert that exon 7 deletion is of the father`s origin, but it is presumptive thanks to electrophoresis results. Moreover, it does not really matter, whether the deletion of the exon 7 is inherited or not, as two detected HACE1 mutations were confirmed to be in trans position. Therefore they were confirmed to cause the disease.

  1. The impact of the presence of various HACE1 transcripts should be discussed in detail.

Unfortunately, there is a little information regarding HACE1 transcripts, however, we know that HACE1 isoform d, that does not contain the sequence of exon 7, as well as the isoform h, that lacks the sequences of exons 6 and 7 are in charge of the proteins formation: 741 aa and 715 aa respectively. Their amplicons are detected in all samples, including references (Figure 5); however, the canonical transcript is present only in healthy individuals but is absent in the proband. Moreover, proband`s deletion of the exon 7 leads to the frameshift and does not allow full-sized protein synthesis due to NMD (Figure 9) (Lines 214-230). We added Supplementary Material 2, where we mentioned all of the discussed isoforms and their structures.

  1. English should be improved.

We have corrected it partly

  1. Low quality of the figures.

We have adjusted the quality of the figures

In conclusion, the manuscript reports new gross deletions in the HACE1 gene. It should be presented as a case report rather than an original paper.   For the original article, a more detailed transcript analysis should be presented. The authors performed a couple of genetic tests, however, the analysis of the results is insufficient.